# Effects of Inhaled Corticosteroids on the Innate Immunological Response to *Pseudomonas aeruginosa* Infection in Patients with COPD

**DOI:** 10.3390/ijms23158127

**Published:** 2022-07-23

**Authors:** Noemi Cerón-Pisa, Hanaa Shafiek, Aina Martín-Medina, Javier Verdú, Elena Jordana-Lluch, Maria Escobar-Salom, Isabel M. Barceló, Carla López-Causapé, Antonio Oliver, Carlos Juan, Amanda Iglesias, Borja G. Cosío

**Affiliations:** 1Instituto de Investigación Sanitaria de Les Illes Balears (IdISBa), Hospital Universitario Son Espases, 07120 Palma de Mallorca, Spain; ceronpisa.n@gmail.com (N.C.-P.); ainamartin89@gmail.com (A.M.-M.); franciscoj.verdu@ssib.es (J.V.); elena.jordana@ssib.es (E.J.-L.); maria.escobarsalom@ssib.es (M.E.-S.); isabelmaria.barcelomunar@ssib.es (I.M.B.); carla.lopez@ssib.es (C.L.-C.); antonio.oliver@ssib.es (A.O.); carlos.juan@ssib.es (C.J.); 2Chest Diseases Department, Faculty of Medicine, Alexandria University, Alexandria 21526, Egypt; whitecoat.med@gmail.com; 3Department of Respiratory Medicine, Hospital Universitario Son Espases, 07120 Palma de Mallorca, Spain; 4Centro de Investigación Biomédica en Red en Enfermedades Infecciosas, Instituto de Salud Carlos III (CIBERINFEC), 28029 Madrid, Spain; 5Department of Microbiology, Hospital Universitario Son Espases, 07120 Palma de Mallorca, Spain; 6Centro de Investigación Biomédica en Red de Enfermedades Respiratorias, Instituto de Salud Carlos III (CIBERES), 28029 Madrid, Spain

**Keywords:** corticosteroids, COPD, *Pseudomonas aeruginosa*, inflammatory cytokines, Toll-like receptors, transcription factors

## Abstract

Inhaled corticosteroids (ICS) use is associated with an increased risk of *Pseudomonas aeruginosa* (PA) infection in patients with COPD. We aimed to evaluate the effects of ICS on alveolar macrophages in response to PA in COPD patients with and without baseline ICS treatment (COPD and COPD + ICS, respectively) as well as smoker and nonsmoker controls. To do so, cells were infected with PA and cotreated with budesonide (BUD) or fluticasone propionate (FLU). The analysis of NF-κB and c-jun activity revealed a significant increase in both factors in response to PA cotreated with BUD/FLU in smokers but not in COPD or COPD + ICS patients when compared with PA infection alone. The expression of Toll-like receptor 2 (TLR2) and the transcription factor c-jun were induced upon PA infection in nonsmokers only. Moreover, in the smoker and COPD groups, there was a significant increase in TLR2 and a decrease in c-jun expression when treated with BUD/FLU after PA infection, which were not observed in COPD + ICS patients. Therefore, the chronic use of ICS seemingly makes the macrophages tolerant to BUD/FLU stimulation compared with those from patients not treated with ICS, promoting an impaired recognition of PA and activity of alveolar macrophages in terms of altered expression of TLR2 and cytokine production, which could explain the increased risk of PA infection in COPD patients under ICS treatment.

## 1. Introduction

Chronic obstructive pulmonary disease (COPD) is characterized by an abnormal inflammatory response of the lungs primarily caused by tobacco smoke that persists even after cessation of smoking [1]. The episodes of acute worsening of the symptoms are known as COPD exacerbations, which commonly occur during the disease process and are associated with higher morbidity and mortality. Therefore, efforts to improve COPD treatment and management require considering both the disease in a broad sense and the exacerbations in particular. Inhaled corticosteroids (ICS) are commonly prescribed to prevent these exacerbations [2]. In contrast to the beneficial effects of ICS, some studies have also shown that its administration is associated with an increased risk of pneumonia in patients with COPD but has not been described in other ICS-treated patients, such as asthmatics, which suggests a specific sensitivity in COPD [3,4]. A recent study showed that higher ICS doses could be a risk factor for *Pseudomonas aeruginosa* (PA) infection in severe COPD patients [5]. Controversy exists as to whether the risk is the same between the different types of ICS used, i.e., budesonide (BUD) or fluticasone propionate (FLU).

The effect of ICS is known to be mediated by binding to the glucocorticoid receptor (GR) located in the cytoplasm of epithelial cells. Once activated, the complex formed by the receptor and the glucocorticoid migrates to the nucleus, where it binds to DNA and directly or indirectly regulates gene transcription. ICS have anti-inflammatory effects, mainly by suppressing the expression of cytokines and promoting the expression of certain chemokines and adhesion molecules in airway epithelial cells (AECs) [6]. On the other hand, Toll-like receptors (TLRs) are a part of the innate immunity recognition system and are expressed under tight regulation in the lung cells involved in front-line host defense, including monocytes, macrophages, and AECs. TLRs recognize microbial components of bacteria, fungi, and viruses, thus mediating a variety of immune responses [7]. For instance, TLR2 is capable of detecting PAMPs (pathogen-associated molecular patterns) in PA [8], and its expression is also known to be induced by this bacterium [9]. Microbial ligands binding to TLR activates intracellular signaling pathways, such as the mitogen-activated protein kinases (MAPK:p38) pathway, and activates nuclear factors such as kappa B (NF-kB) and c-jun, which regulate the production of inflammation-related mediators such as interleukine-8 (IL-8), interleukine-10 (IL-10), and tumor necrosis factor-alpha (TNF-α) [10] in order to control infection [11]. Of note, the proinflammatory IL-8 is known to be increased in COPD patients, especially during exacerbations [12].

Macrophages and neutrophils are essential for the correct response of the immune system against infection since they have multiple mechanisms for the recognition and subsequent elimination of pathogens. Despite the increase in the number of macrophages and neutrophils in the airways of patients with COPD, respiratory infections are common, suggesting that the function of the immune system is affected [13]. The effects of ICS on macrophages are complex. Recently, we have shown in our laboratory that *Haemophilus influenzae* infection decreases the anti-inflammatory properties of the corticosteroids through a decrease in histone deacetylase (HDAC) activity, which may contribute to the success of infection [14].

Given all these data and the current knowledge suggesting that ICS could be associated with a higher risk of PA infection, we hypothesized that ICS could alter the innate immune response to PA infection in COPD patients. To answer this question, we first characterized the inflammatory cytokine profile in serum and bronchoalveolar lavage (BAL) from COPD patients versus COPD patients under long-term treatment with ICS in comparison to smoker and nonsmoker controls. We then studied the inflammatory response in alveolar macrophages isolated from these groups upon treatment with PA alone or together with ICS (either BUD or FLU) to analyze the expression of TLR2 and c-jun and the activity of c-jun and NF-kB.

## 2. Results

### 2.1. Patient Characteristics

Table 1 shows a summary of the baseline and clinical characteristics of the studied subjects. Forty-nine subjects were recruited and distributed into four groups: COPD treated with ICS (COPD + ICS; *n* = 11, 22.4%), COPD not treated with ICS (COPD; *n* = 16, 32.7%), smokers (N; *n* = 12, 24.5%) and nonsmokers (NS; *n* = 10, 20.4%) with normal lung function. The majority of COPD patients and the smokers group were men, while the nonsmokers were mainly females (*p* = 0.018, Table 1). The FEV1 (% predicted), FEV1/FVC%, and KCO were significantly reduced in COPD groups compared with smokers and nonsmokers (*p* < 0.05, Table 1). According to GOLD stages, 8.2% and 9.1% of COPD + ICS were stage IV and III, respectively, while 6.25% of the COPD group were stage III (*p* = 0.389, Table 1). Among the COPD + ICS group, seven patients (63.6%) were using fluticasone propionate 500 μg twice daily (1000 μg/day), three patients (27.3%) were using beclomethasone 100 μg once daily, and two patients (18.2%) were using budesonide 320 μg twice daily (640 μg/day).

### 2.2. Inflammatory Pattern in Patients’ BAL and Serum

To study the effect of ICS baseline treatment on inflammation in the absence of PA infection, the levels of the key anti-inflammatory cytokine IL-10 and proinflammatory cytokine IL-8 were determined in both BAL and serum of COPD + ICS, COPD, smokers (N), and nonsmokers (NS). In BAL, IL-10 was significantly decreased, whereas IL-8 was significantly increased in both COPD and COPD + ICS groups when compared with smokers and nonsmokers (*p* < 0.05, Figure 1A). Interestingly, the differences were higher in the COPD + ICS group for IL-10 (*p* < 0.005) (Figure 1A). In serum, IL-8 followed the same trend as in BAL, thus being significantly higher in both COPD and COPD + ICS versus nonsmokers and smokers (Figure 1B). Opposite to BAL, serum IL-10 was significantly increased in both COPD groups compared with smokers and only in COPD + ICS when compared with nonsmokers; and IL-10 levels were significantly higher in COPD + ICS compared with COPD (*p* < 0.05, Figure 1B). TNF-α was also analyzed, but no statistically significant differences were raised either in serum or in BAL (data not shown).

### 2.3. TLR2 and c-Jun Gene Expression in Primary Human Alveolar Macrophages upon Infection and in Response to ICS Cotreatment

Induction of TLR2 and c-jun happens regularly during infections and can also be triggered by corticosteroids such as BUD and FLU [8]. Given that alveolar macrophages are essential for the management of infections, especially through the activation of the TLR pathway and their effectors, the expression of TLR2 and c-jun was assessed in primary human alveolar macrophages isolated from nonsmokers (NS), COPD patients (COPD), or COPD patients under ICS treatment (COPD + ICS) that were infected in vitro with PA and further treated with BUD or FLU. Treatments with BUD or FLU alone as well as infection with PA alone were used as controls. Alveolar macrophages from the nonsmokers group were the only ones to significantly increase TLR2 mRNA levels in response to PA infection (C vs. CP *p* < 0.05, Figure 2A), whereas they remained unchanged in smokers, COPD, and COPD + ICS groups. The expression of TLR2 was also triggered by BUD and FLU in nonsmokers but not in smokers, and it was upregulated only in response to BUD (but not by FLU) in COPD patients, whereas no statistically significant effect was observed in COPD + ICS patients. Despite that TLR2 levels were not induced by PA either in smokers or COPD patients, BUD and FLU cotreatment during PA infection rescued the response to PA increasing TLR2 levels in the same groups of patients (*p* < 0.05, Figure 2A). Interestingly, none of the corticosteroid treatments or their combinations during PA infection was capable of inducing TLR2 expression in COPD + ICS group (*p* > 0.05, Figure 2A). On the other hand, c-jun expression was shown to be significantly increased in macrophages after PA infection in both nonsmokers and smokers but not in COPD or COPD + ICS individuals (*p* < 0.05, Figure 2B). Of note, the levels of c-jun decreased after PA infection plus cotreatment with BUD and FLU in the nonsmokers group as well as in smokers and COPD patients (*p* < 0.05, Figure 2B), thus indicating a good response to these drugs in terms of reducing inflammation, but this response was not found in macrophages from COPD + ICS subjects (*p* > 0.05, Figure 2B). Taken together, these results suggest that the TLR2 and c-jun-mediated immune responses toward PA infection are impaired in alveolar macrophages from COPD subjects, and although treatment with corticosteroids FLU and BUD may rescue the response in COPD, these drugs have no effect in COPD individuals under ICS baseline treatment.

Next, the activity of the inflammatory mediators NF-κB and c-jun in response to ICS (BUD and FLU) was assessed in the context of PA infection using the same experimental design as above. The nuclear extracts from primary alveolar macrophages, NF-κB, and c-jun activities, although not statistically significant, appear increased upon PA stimulation only in nonsmokers but not in smokers or COPD patients (Figure 3A for NF-κB, and 3B for c-jun). However, NF-κB activity significantly increased after PA infection plus cotreatment with BUD in the smokers group in comparison to PA alone, and a similar trend was observed for FLU (Figure 3A). Further, the activity of c-jun was significantly increased upon cotreatment with PA and BUD or FLU among the COPD group and smokers, respectively, in comparison to PA alone (*p* < 0.05, Figure 3B), whereas no response was observed in the COPD + ICS group (Figure 3B). In summary, a modest increase in the activity of NF-κB and c-jun transcription factors was observed in the alveolar macrophages of non-smokers in response to PA infection, whereas the response is blunted or even follows the opposite trend in smokers and COPD individuals. Moreover, while alveolar macrophages in smokers and COPD subjects responded to some extent to cotreatment with BUD/FLU, those from COPD patients under ICS were apparently not responsive to these drugs.

## 3. Discussion

In this study, we analyzed the effects of ICS on the innate immunological response with a focus on the general inflammatory response and behavior of the alveolar macrophages in COPD patients. We observed an overexpression of TLR2 in the alveolar macrophages in COPD patients after costimulation with PA infection and the corticosteroids BUD and FLU, which was associated with higher NF-κB activity; however, this response was not observed in the COPD patients that were under ICS baseline treatment. Therefore, the long-term treatment of COPD patients with ICS might have blunted the normal response that is orchestrated by alveolar macrophages after PA infection, both in terms of TLR2 expression and effective cytokine production, impairing the management of the infectious process. This observation supports the finding in the clinical studies of Eklöf et al. [15] and Shafiek et al. [5], who found that high doses of ICS could be a risk factor for PA in severe COPD patients.

In BAL, IL-10 was significantly decreased, whereas IL-8 was significantly increased in both COPD and COPD + ICS groups when compared with both smokers and nonsmokers (*p* < 0.05). Interestingly, the differences were higher in the COPD + ICS group for IL-8 (*p* < 0.005, Figure 1A). In serum, IL-8 followed the same trend as in BAL, thus being significantly higher in both COPD and COPD + ICS versus nonsmokers and smokers (Figure 1B). Opposite to BAL, serum IL-10 was significantly increased in both COPD groups compared with smokers and only in COPD + ICS when compared with nonsmokers and, of note, IL-10 levels were significantly higher in COPD + ICS compared with COPD (*p* < 0.05, Figure 1B). TNF-α was also analyzed, but no statistically significant differences were raised either in serum or in BAL (data not shown). Taken together, these results suggest that COPD patients, with and without ICS treatment, show a sharped upregulation of proinflammatory IL-8 levels in BAL and serum. In addition, the levels of the anti-inflammatory IL-10 cytokine are either down- or upregulated depending on the sample (down in BAL and up in serum), but when compared with nonsmokers, serum IL-10 was upregulated specifically in the COPD + ICS.

Alveolar macrophages are essential for host defense through their ability to detect pathogens that attack the airways and to regulate both innate and adaptive immunity [16]. ICS mimic the effects of endogenous corticosteroids, normally resulting in a decrease in proinflammatory mediators by alveolar macrophages [17] upon exposure to microbial or oxidative stress triggers [18]. For instance, IL-10 is an important mediator in the resolution of lung inflammation. Peñazola et al. showed that IL-10 deficient mice had a higher mortality rate, greater expression of proinflammatory cytokines, and higher recruitment of neutrophils to the lungs when infected by *Streptococcus pneumoniae* [19]. Furthermore, Jiang et al. found a reduction in serum IL-10 levels in COPD patients that was negatively correlated with the severity of the disease [20]. We found a reduction in IL-10 levels in BAL from COPD compared with smokers and nonsmokers regardless of the ICS treatment. On the contrary, serum levels of IL-10 were upregulated in both COPD and COPD + ICS compared with nonsmokers, but only in the ICS group when compared with smokers. It is possible that the inhibition of IL-10 production in BAL from COPD patients caused by corticosteroids may hinder the resolution of inflammation within the lung (therefore supporting the upregulation of IL-8 in BAL), whereas, systemically, IL-10 levels would be upregulated instead, especially in patients under ICS treatment. In addition, we showed that BAL and serum IL-8 levels followed the same trend, thus being increased in both COPD and COPD + ICS groups compared with nonsmokers and smokers, denoting that ICS are not promoting a reduction in the proinflammatory IL-8 release. In COPD patients, IL-8 appears to be massively secreted from alveolar macrophages and neutrophils and tends to be higher in the serum and airways [21]. Nightingale et al. found that high doses of ICS failed to suppress neutrophilic inflammation induced by ozone inhalation in normal subjects [22] to a magnitude similar to that found in COPD, which could explain the failure of IL-8 and TNF-α suppression [23]. This could be explained on the basis of the chronic use of ICS that could suppress the bacterial-induced release of proinflammatory IL-6 but not that of IL-8, as previously demonstrated [24].

The role of TLR2 in the pathogenesis of COPD is controversial, with conflicting findings in the literature [25,26,27,28]. These conflicting data are likely due to differences between the cells studied (e.g., peripheral blood monocytes vs. macrophages), different patient cohorts, potential tissue specificity of TLR expression, and the experimental models used (e.g., acute versus chronic exposure) [18]. We observed that TLR2 was induced in alveolar macrophages from nonsmokers in response to PA infection but remained unchanged in smokers as well as in COPD and COPD + ICS patients, which might indicate a common situation in these individuals in which the TLR2 response to infection is compromised. However, PA-infected alveolar macrophages showed overexpression of TLR2 when treated with BUD or FLU in the COPD group as well as in smokers, denoting a similar inflammatory response in both groups. Our results are similar to those found by Provost K. et al., although they used *H. influenzae* and *S. pneumoniae* [29]. They found that bacterial infection induced a reduction in TLR expression (TLR2 and 4) that was restored after BUD and FLU treatment in COPD patients and smokers without COPD [29]. Similarly, Ji et al. [30] and von Scheele et al. [27] found that subsequent stimulation by lipopolysaccharides (the main component of Gram-negative bacterial envelopes) [31] and BUD was associated with TLR2 overexpression. This could be explained on the basis that smoking exposure is associated with modulation of the inflammatory response and alteration of the response of macrophages in infection, including phagocytosis of microbes [32,33,34]. On the other hand, we did not observe a further surge in TLR2 overexpression in the COPD + ICS group either after PA infection or with BUD/FLU treatment when compared with its control. The maintained overexpression of TLR2 mRNA in the COPD + ICS group may be due to tolerance of TLR2 that could be explained by continuous and repeated stimulation by chronic ICS use in a manner similar to the experiment from Lea et al. [35]. Our results support the concept that long-term use of ICS in COPD is associated with an altered inflammatory response to infection, especially PA infection.

NF-κB and c-jun are common transcription factors involved in the inflammatory response. TLRs recruit adaptor proteins MyD88 and TRIF/TICAM1, which transmit the signal to downstream kinases as ubiquitin ligases involved in the degradation of NF-κB inhibitors. The result is an activation of transcription factors from the NF-κB and AP-1 (also called c-jun) families [36]. In our setup, nonsmokers showed a trend of NF-κB and c-jun upregulation in activity in response to PA, which is maintained in cotreatment with BUD or FLU. On the other hand, NF-κB and c-jun activity in the nucleus increased with either BUD and FLU treatment compared with PA stimulation alone in smokers (both factors) and COPD not treated with ICS (for c-jun) groups, but not in the COPD + ICS subjects, which was consistent with the increase we observed in TLR2 expression. Our results are in accordance with Rossios et al., who showed that glucocorticoids directly inhibit NF-κB and c-jun activity [30]. Similarly, Gagliardo et al. found that FLU inhibits IKK-driven NF-KB activity in COPD and smokers [37]. Given that, contrary to its activity, we found a downregulation of mRNA levels of c-jun in the COPD group in response to PA and BUD or FLU cotreatment, which could be due to post-transcriptional changes affecting the translocation of c-jun to the nucleus of the cell. In the COPD + ICS group, the activity of NF-κB and c-jun were similar in all conditions, which could be explained by the lack of overexpression of TLR2 or c-jun. Taken together, our results indicate that the TLR2 and c-jun-mediated immune response toward PA infection is diminished in alveolar macrophages from COPD subjects, and although treatment with corticosteroid drugs (either FLU or BUD) can rescue the response in COPD, they do not have any effect in COPD individuals who are under ICS baseline treatment.

Our study has some strengths and limitations. On the one hand, this is the first study to explore the inflammatory response to PA infection in alveolar macrophages from a well-characterized COPD population treated with ICS. On the other hand, although the sample is large enough to demonstrate changes in some of the inflammatory readouts, a larger size may be required to explore the role of alternative pathways. Given the exploratory nature of the experiments, we can speculate but not translate our findings into clinical outcomes. Further, we could not study the relationship between the macrophage response and the disease severity, as only 15% of COPD patients were in stage III–IV, which is quite an insufficient number for study. Although modest, this study reveals, for the first time, alterations in the immune response to PA infection in COPD patients that are being treated with ICS. These findings open a gate to designing future experiments in order to improve the management of PA infections in COPD patients.

## 4. Materials and Methods

### 4.1. Study Design and Ethics

This is a cross-sectional, descriptive, comparative, and controlled study. All the participants signed their informed consent after being aware of the nature and objectives of the study. The project was approved by the Ethics Committee of the Balearic Islands in Spain (lB 3537/17 PI, 20 December 2017).

### 4.2. Population

A total of 27 subjects with a diagnosis of COPD according to international guidelines [1] were separated into 11 COPD patients on ICS treatment (COPD + ICS) for at least 6 months and 16 COPD patients on long-acting bronchodilators without ICS (COPD). In addition, 12 smokers and 10 nonsmoker patients without evidence of airway obstruction on spirometry undergoing bronchoscopy for clinical reasons were included as hemoptysis, lung nodules, lung mass, or atelectasis, whereas the BAL was taken from the healthy lung. Patients with active inflammatory diseases, infection for *Pseudomonas aeruginosa* in bronchoalveolar aspirate (BAS), receiving treatment with antibiotics, immunosuppressive agents, or patients with a COPD exacerbation within the previous 6 weeks were excluded. All participants underwent forced spirometry following international guidelines [38] using the reference values of the Mediterranean population [39]. According to the FEV_1_, the COPD patients were classified based on Global Initiative in Obstructive Lung Diseases (GOLD) categories as mild (FEV_1_ ≥ 80%), moderate (FEV_1_ ≥ 50–<80%), severe (FEV_1_ ≥ 30–<50%), and very severe (FEV_1_ < 30%) [1]. A detailed history of ICS therapy from COPD + ICS patients regarding the type of ICS and dosage was taken. Fiberoptic bronchoscopy was performed, and BAL was obtained from the right middle lobe or the lingula as previously described [40] and from the healthy lung of the participants. Alveolar macrophages were isolated from the BAL as previously described, incubated in 24-well plates, and cultured at 37 °C in a humidified atmosphere with 5 % CO_2_ in RPMI 1640 medium containing 0.5 % fetal calf serum (FCS), 5 % HEPES, and supplemented with antibiotics (50 U/mL penicillin and 50 Ag/mL streptomycin) for 24 h [14].

### 4.3. Bacterial Strains and Preparation

*Pseudomonas aeruginosa* strain PAO1 was grown overnight in LB (Luria Bertani) broth at 37 °C and 180 rpm agitation [41].

### 4.4. Peripheral Blood Sampling and Processing

Blood samples (8 mL) were obtained from all participants by peripheral venipuncture and collected in nonheparinized tubes. Blood was allowed to clot and centrifuged for 10 min at 3000 rpm at 4 °C. Then, serum was aliquoted and stored at −80 °C for further analysis.

### 4.5. Cytokine Determination

The concentrations of IL-10 and IL-8 in both the serum and the supernatant of BAL were analyzed using a human cytokine magnetic bead panel (Merck Millipore, Billerica, MA, USA) according to the manufacturer’s instructions. A total of 25 μL of each sample, 25 μL of assay buffer, and 25 μL of premixed beads were added to the sample wells. The plate was incubated with shaking overnight (16–18 h) at 4 °C. The next day, it was washed three times, and 50 μL of detection antibodies were added to each well. Samples were incubated with shaking on a vibrating plate for 1 h at room temperature, and afterward, 50 μL of streptavidin–phycoerythrin were added to each well containing the 50 μL of detection antibodies. Samples were then incubated with shaking on a vibrating plate for 30 min at room temperature. Samples were finally washed three times, 150 μL of pLUS driving liquid was added to all wells, and the plate was read in a MAGPIX^®^ (Luminex, Austin, TX, USA). Assay sensitivity according to the manufacturer was as follows: 0.56 pg/mL for IL-10, 0.11 pg/mL for IL-6, 0.13 pg/mL for IL-8, and 0.16 pg/mL for TNF-α.

### 4.6. Cell Culture

Primary human alveolar macrophage cell cultures were infected with PAO1 strain at a multiplicity of infection (MOI) of 100:1 [42] for 3 h with the simultaneous addition of FLU (10^−6^ M) (Cayman Chemical, Ann Arbor, MI, USA) or BUD (10^−6^ M) (Cayman Chemical, Ann Arbor, MI, USA) when indicated, concentrations known to be effective from our previous experiments (10). Following treatment, macrophages were isolated and RNA extracted using Trizol for real-time PCR (see below), and nuclear proteins were extracted using a nuclear extract kit^®^ (ActiveMotif, Carlsbad, CA, USA) and kept at −80 °C.

### 4.7. NF-κB and c-Jun Activity

NF-κB and c-jun activation were assessed in the nuclear protein extracts of alveolar macrophages using the TransAM NF-κB p65 Transcription Factor Assay Kit and c-jun Assay Kit (Active Motif, Carlsbad, CA, USA) as previously described [14].

### 4.8. RT-PCR

Total RNA from cultured primary alveolar macrophages was isolated using TRIzol Reagent (GIBCO-BRL Life Technologies, Waltham, MA, USA) following the manufacturer’s instructions. A total of 1 μg of RNA was reverse-transcribed into cDNA using the SensiFAST cDNA Synthesis Kit (Bioline, London, UK) as previously described [43]. Subsequent real-time PCR reactions were performed in duplicate (CFX96 Real-Time System, C1000 Thermal Cycler, Bio-Rad, Hercules, CA, USA) using the AMPLIFYME SYBR Universal Kit (Blirt SA, Gdańsk, Poland) following the manufacturer’s instructions. The thermocycling conditions were as follows: 95 °C for 3 min, followed by 40 cycles of denaturation at 95 °C for 30 s, annealing at 60 °C for 30 s, and extension at 72 °C for 30 s. Product specificity was confirmed in initial experiments by melting curve analysis, and PCR efficiency was calculated for each specific primer. The primers to determine the expression of each gene are shown in Table 2. The expression of the housekeeping gene codifying for the β2-microglobulin protein (B2M) was used as an internal control to normalize the expression of each target gene. Relative quantification with the 2^−ΔΔCq^ method [44] was used to evaluate the relative expression of mRNAs of interest.

### 4.9. Statistical Analysis

All the data were presented as mean ± standard deviation (SD) or number (*n*) and percentage (%) as appropriate. Comparison between experimental groups was performed using analysis of variance (one-way ANOVA) followed by Bonferroni post hoc tests to compare different groups when the analysis was carried out to observe the effect of the treatment on each individual group (mRNA expression), or two-way ANOVA followed by either Bonferroni or Dunnet’s post hoc test when carried out to assess the effect among the groups. A *p*-value of <0.05 was considered to represent a statistically significant difference. All statistical analyses were carried out using GraphPad InStat version 8.00 (GraphPad Software, Inc, La Jolla, CA, USA).

## 5. Conclusions

The blunted response to PA in COPD patients treated with ICS may translate into an absence of recognition of PA by the alveolar macrophages as being detected by a decreased expression of TLR2 and c-jun, which could be one of the mechanisms explaining the increased risk of PA infection in COPD patients. Further, the chronic use of ICS in these patients could induce tolerance of their macrophages’ TLRs and explains the lack of response to BUD/FLU stimulation when compared with COPD patients not using ICS.

## Figures and Tables

**Figure 1 ijms-23-08127-f001:**
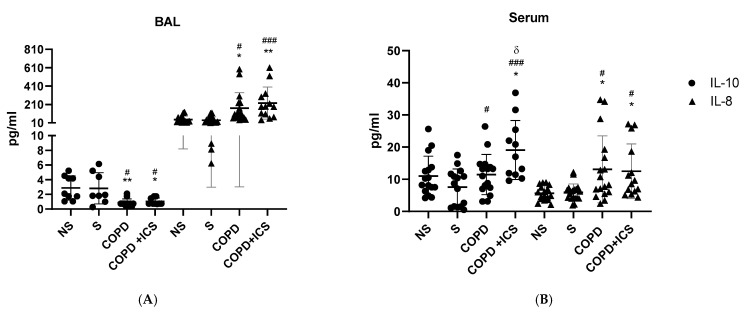
Cytokine expression in BAL (**A**) and in serum (**B**) of the studied groups analyzed by ELISA: NS, nonsmoker; COPD, chronic obstructive pulmonary disease without ICS baseline treatment; COPD + ICS, chronic obstructive pulmonary disease with inhaled corticosteroids use; IL, interleukin. * *p* < 0.05 and ** *p* < 0.01 vs. NS; # *p* < 0.05, and ### *p* < 0.005 vs. S; δ *p* < 0.05 vs. COPD. ● IL-10; ▲ IL-8.

**Figure 2 ijms-23-08127-f002:**
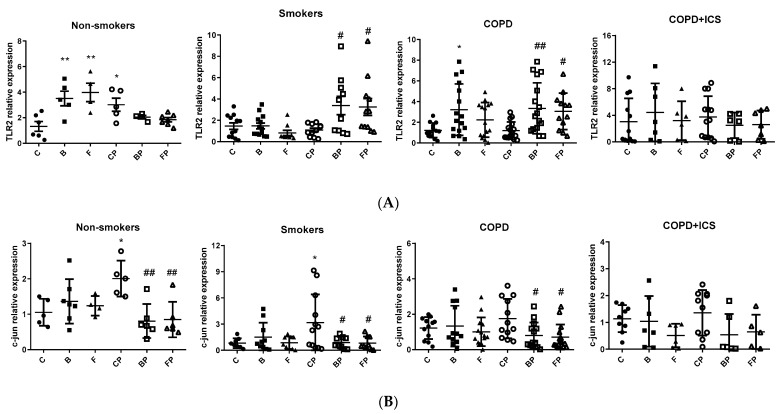
TLR2 (**A**) and c-jun (**B**) mRNA expression in alveolar macrophages from nonsmokers, smokers, COPD, and COPD patients under corticoid treatment (COPD + ICS): C, control; B, stimulated with BUD; F, stimulated with FLU; CP, stimulated with *Pseudomonas aeruginosa*; BP, stimulated with *Pseudomonas aeruginosa* and BUD; FP, stimulated with *Pseudomonas aeruginosa* and FLU. One-way ANOVA, Bonferroni’s post hoc; * *p* < 0.05 vs. C; ** *p* < 0.01 vs. C; # *p* < 0.05 and ## *p* < 0.01 vs. CP. C: ●; B: ■; F:▲; CP: ○; BP: □; FP: Δ.

**Figure 3 ijms-23-08127-f003:**
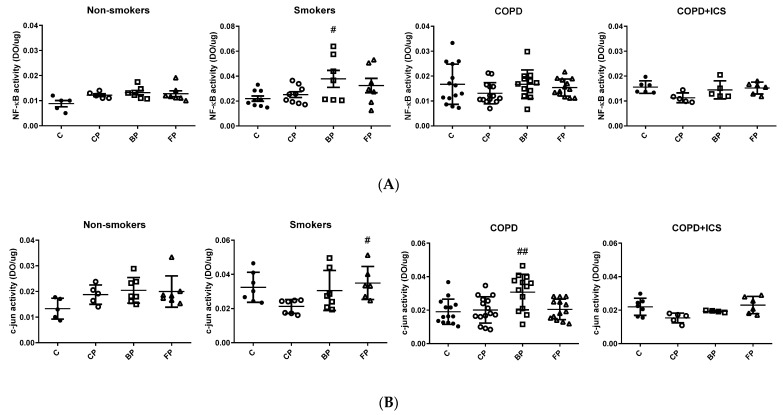
NF-κB (**A**) and c-jun (**B**) activity in alveolar macrophages: NS, nonsmoker; COPD, chronic obstructive pulmonary disease; COPD + ICS, chronic obstructive pulmonary disease with inhaled corticosteroids use; C, control; CP, stimulated with *Pseudomonas aeruginosa*; BP, stimulated with *Pseudomonas aeruginosa* and budesonide (BUD); FP, stimulated with *Pseudomonas aeruginosa* and fluticasone propionate (FLU). *n* = 6–8 per group; two-way ANOVA, Bonferroni–s post hoc; # *p* < 0.05 and ## *p* < 0.01 vs. CP. C: ●; CP: ○; BP: □; FP: Δ.

**Table 1 ijms-23-08127-t001:** The baseline characteristics of the studied population.

Variable	COPD + ICS (*n* = 11)	COPD (*n* = 16)	NS (*n* = 10)	S (*n* = 12)	Sig. (*p*)
Age	68.27 ± 7.35	67.25 ± 8.23	61.8 ± 13.89	64 ± 9.85	0.392
Gender Male/female	7 (63.6)/4 (36.4)	13 (81.3)/3 (18.8)	2 (20)/8 (80)	8 (66.7)/4 (33.3)	**0.018 ***
Smoking history Smoker Exsmoker Nonsmoker	4 (36.4) 7 (63.6) 0 (0)	9 (56.3) 7 (43.8) 0 (0)	0 (0) 0 (0) 10 (100)	8 (66.7) 4 (33.3) 0 (0)	**<0.001 ***
Smoking index	61 (40–69)	52.5 (40–60)	0 (0–0)	39.5 (23–60)	**<0.001 ***
BMI (kg/m^2^)	25.33 ± 7.09	28.05 ± 4.02	25.53 ± 4.18	25.9 ± 4.75	0.583
**Spirometry**					
FVC (L)	3.27 ± 0.88	3.45 ± 0.84	3.05 ± 0.59	4.09 ± 1.13	0.300
FVC (% predicted)	86.8 ± 11.44	95.43 ± 23.72	100.25 ± 2.87	100.6 ± 11.76	0.404
FEV_1_ (L)	1.74 ± 0.69	2.08 ± 0.52	2.35 ± 0.65	3.01 ± 0.85	**0.014 ***
FEV_1_ (% predicted)	68 (50–72)	71.25 (60–90)	99 (94–103)	102 (98–105)	**0.005 ***
FEV_1_/FVC	56.56 (48–66.88)	62.26 (55.96–65.52)	75.32 (70.45–81.3)	73.4 (70–76)	**0.001 ***
DLCO	62 (57–67)	69 (58–82)	83 (82.5–85.5)	83.5 (71.5–95.5)	0.118
KCO	69.5 (54–83)	64 (57.5–77.5)	88 (87–90.5)	80 (77–81)	**0.033 ***
**GOLD categories ^$^**					
GOLD I	2 (18.2)	4 (25)	NA	NA	0.389
GOLD II	5 (45.5)	8 (50)			
GOLD III	1 (9.1)	1 (6.25)			
GOLD IV	2 (18.2)	0 (0)			

Abbreviations: ICS, inhaled corticosteroids; N, smoker; NS, nonsmoker; BMI, body mass index; *p*/y, packs/year index; FVC, forced vital capacity; FEV1, forced expiratory volume in 1 s; DLCO, diffusion capacity for carbon monoxide; KCO, the carbon monoxide transfer coefficiency; BDT, postbronchodilator test; GOLD, Global Organization for Pulmonary Disease; NA, not assessed. * Significant *p* < 0.05 refers to the comparison between four groups. ^$^ One missing data in COPD + ICS and three missing data in COPD.

**Table 2 ijms-23-08127-t002:** Primers used in this study.

Gene	Sequence
TLR2	F: GGACTTCTCCCATTTCCGTCT
	R: CTCCAGGTAGGTCTTGGTGTTC
c-jun	F: AAAGGATAGTGCGATGTTTC
	R: TAAAATCTGCCACCAATTCC
B2M	F: ACCCCCACTGAAAAAGATGAG
	R: ATCTTCAAACCTCCATGATGC

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
