# Peer review of "Effects of Inhaled Corticosteroids on the Innate Immunological Response to Pseudomonas aeruginosa Infection in Patients with COPD"

_ijms, 2022, doi:10.3390/ijms23158127_

Round 1
Reviewer 1 Report
Thank you for inviting me to review this manuscript: Effects of inhaled corticosteroids on the innate immunological response to Pseudomonas aeruginosa infection in patients with COPD. The authors have done a good job of investigating the inflammatory response to PA infection in alveolar macrophages from a population of COPD patients treated with ICS.
Studies like these are important to find out how we can avoid severe infections in patients with COPD.
I have a few comments.
1. A recent study showed that higher ICS doses could be a risk factor for Pseudomonas aeruginosa (PA) infection in severe COPD patients. Also worth mentioning is a large epidemiologist study: Josefin Eklöf et. al 2021 Thorax. Use of inhaled corticosteroids and risk of acquiring Pseudomonas aeruginosa in patients with chronic obstructive pulmonary disease
2. I do not understand this calculation:
According to the Global Organization for Pulmonary Disease (GOLD), forty percent of the COPD group were GOLD stage II, while 15% were GOLD stage III-IV. What about the last 45%:
3. Although it is a cross-sectional study and exploratory study, there are very few patients so it may have had an impact on the outcome.
Reviewer 2 Report
This study aims to investigate whether inhaled corticosteroids could alter the innate immunological response to pseudomonas aeruginosa (PA) infection in patients with COPD.
To this aim the authors examined 11 COPD subjects under ICS treatment (COPD + ICS), 16 COPD subjects without ICS treatment (COPD) and 10 nonsmokers with normal lung function. The levels of TNF alpha, IL-6, IL-8 e IL-10 were examined in serum and BAL supernatant of the 3 groups of subjects. Moreover, macrophage cultures were obtained from the 3 groups of subjects and were infected with PA and then treated with steroids (budesonide or fluticasone). In nonsmoking subjects, gene expression of TLR2 increased with PA infection and persisted at high level after steroid treatment. In COPD subjects TLR2 expression did not increased with PA infection, but increased with steroid treatment. No significant changes in TLR2 expression were observed with PA infections or steroid treatment in macrophage cultures obtained from COPD + ICS. C-jun expression increased in nonsmoking subjects with PA infection and decreased with steroid treatment. No significant changes were observed in C-jun expression with PA infections or steroid treatment in macrophage cultures obtained from COPD and COPD + ICS. When the activity of the transcription factors NF-kB and c-jun was examined in the nuclear extracts of alveolar macrophages the only significant changes was an increased activity of both factors after steroid treatment in the COPD group. Based on these results the authors concluded that the chronic use of ICS makes the macrophages tolerant to steroids promoting an impaired response to PA which could explain the increased risk of PA infection in COPD patients under ICS treatment.
There are some concerns with this manuscript
1) The design of the study is confusing and in particular, it is difficult to determine whether this research was based on a clear plan or whether different inflammatory parameters were correlated in the hope that some significant findings would turn up. Moreover, analysis of BAL and serum, as presented, are poorly related to those of macrophage cultures and it seems that two different studies were merely combined without an analysis plan.
2) The lack of information (in the Introduction) on the relevance for the present study of TNF alpha, IL-6, IL-8 and IL-10 does not help to clarify the design of the study.
3) Conclusions, as reported in the last paragraph of the manuscript, are not completely supported by the results or at least it is not clear which are the results that support the conclusions. In which sets of experiments a decreased expression of TLR2 and c-jun was observed?
4) It would be of interest to investigate the effect of steroid treatment on macrophage cultures in the absence of PA infection.
5) It would be of interest also to investigate a group of smoking subjects without COPD, to explore whether smoking may have affected the results.
6) More information should be provided on the clinical history of nonsmoking subjects. Why did they undergo bronchoscopy?
7) Details on ICS therapy and GOLD categorization in the COPD+ICS group should be included
8) Was there any relation between macrophage response and disease severity?
9) Presentation of results in figures 1-5 as individual values and medians instead of means and SD would allow to interpret variability. Considering the small number of subjects examined in this study, this appears feasible.
10) A lot of comments (including one conclusion) are reported in the “Results”. The majority of them should be transferred to the “Discussion”
11) Table 2: it is not clear to which comparison the p value refers (considering that table 2 includes 3 groups).
12) Table 1: what is B2M?
13) Introduction: the last paragraph (lines 81-84) should be deleted from the introduction because it refers to the results of the study.
14) Abstract: “The analysis of NF-KB and c-jun activity revealed a significant increase in both factors in response to PA in COPD” Figures 5 (a and b) does not show increase in NF-KB and c-jun activity in response to PA in COPD
15) Discussion:
- lines 280-296: the paragraph on IL-10 is confusing and very difficult to follow for the reader
- lines 322-323: ”That was already higher than the non-ICS treated COPD or the non-smokers controls”. I did not find this result in the “Results”. Is there an explanation for this finding?
- lines 332-334: it is not clear to which results this sentence refers
- lines 337-338:”we found dowregulation of mRNA levels of c-jun in the COPD group in response to PA and BUD or FLU co-treatment”: no downregulation is shown in figure 3B
Reviewer 3 Report
The report from Ceron-Pisa et al. discusses the effects of inhaled corticosteroids (ICS) on cytokine levels (IL-6/8/10/TNFa) in BAL and serum of COPD and non-COPD patients, and their effects on alveolar macrophage TLR2 and c-jun transcript expression during bacterial infection (PA). The manuscript is generally well written but the data and their analysis are often not convincing, mainly because of experimental bias.
Major comments:
1. The recruitment of patients is not properly adequate leading to several bias that preclude a correct interpretation of the results: (i) non-smokers are selected for the non-COPD group while current or ex-smokers would be appropriate; (ii) non-smokers are mainly female while COPD and COPD+ICS are mainly male; (iii) the pulmonary function in the COPD group with ICS is lower than in the COPD group without ICS. In addition, it would be useful to present the levels of severity for each group in Table 2. Finally, for how long were the patients on ICS in the COPD group?
2. Fig 1A: although non-significant statistically, IL-8 levels are increased in COPD patients without ICS so the conclusion should apply to both COPD groups.
3. Fig 2 to 4: means should be represented otherwise the plots are not sufficiently informative. The main issue here is the fact that the basal expressions of TLR2 (5 times more than the 2 other groups) and c-jun are already deregulated in COPD+ICS patients therefore the additional treatments of BUD or FLU are ineffective. The results may all lead to the conclusion that the response to PA is different in COPD+ICS patients solely because the inflammatory response is already altered.
4. The findings are very often not convincing because of the dispersion of the data: (i) Fig1A, COPD and COPD+ICS for IL8 (+/- 200pg/ml); (ii) Fig 2A/B, huge SD; (iii) Fig5, the folds are not much different and the comparative group is not appropriate (it should be CP and not C).
Minor comments:
1. The organization of the figures is not optimal since the reader has to navigate between Fig2/3/4
2. Additional details could be added in Materials and Methods: (i) lines 121-125: what are the volumes of samples, incubation time, temperature, etc. (ii) line 131: when were the cells isolated; (iii) table 1: what is the reference gene? Etc.
3. There are a few typos and some English editing to be done.
Round 2
Reviewer 2 Report
The response to my comments is satisfactory
Author Response
We thank the reviewer for considering that we have adequately addressed his/her revision and we have revised the whole manuscript for English spells.
Reviewer 3 Report
The authors have adequately answered.
I still have 3 remarks:
- The statistical signs are not always defined in the figure legends (## for Fig2; #; and ## for Fig3)
- Format small or capital letters in figure panels.
- Fig3 should be separated into 8 panels as in Fig2
